# Research on the Mechanical and Thermal Properties of Carbon-Fiber-Reinforced Rubber Based on a Finite Element Simulation

**DOI:** 10.3390/polym16152120

**Published:** 2024-07-25

**Authors:** Xiaocui Yang, Xinmin Shen, Wenqiang Peng, Daochun Hu, Xiaoyong Wang, Haichao Song, Rongxing Zhao, Chunmei Zhang

**Affiliations:** 1Engineering Training Center, Nanjing Vocational University of Industry Technology, Nanjing 210023, China; 2019101052@niit.edu.cn (X.Y.); 2018100921@niit.edu.cn (D.H.); 1996100156@niit.edu.cn (X.W.); 2002100250@niit.edu.cn (H.S.); 2019100986@niit.edu.cn (R.Z.); 2019300976@niit.edu.cn (C.Z.); 2Field Engineering College, Army Engineering University of PLA, Nanjing 210007, China; 3College of Aerospace Science and Engineering, National University of Defense Technology, Changsha 410073, China

**Keywords:** carbon-fiber-reinforced rubber, mechanical property, thermal property, finite element simulation, repeatable unit cell, rule of mixture, fiber volume fraction

## Abstract

The comprehensive performance of rubber products could be significantly improved by the addition of functional fillers. To improve research efficiency and decrease the experimental cost, the mechanical and thermal properties of carbon-fiber-reinforced rubber were investigated using finite element simulations and theoretical modeling. The simplified micromechanical model was constructed through the repeatable unit cell with periodic boundary conditions, and the corresponding theoretical models were built based on the rule of mixture (ROM), which can be treated as the mutual verification. The simulation results suggest that, in addition to the fiber volume fraction Vf_c_ increasing from 10% to 70%, the longitudinal Young’s modulus, transversal Young’s modulus, in-plane shear modulus, longitudinal thermal expansion coefficient, and transversal thermal expansion coefficient changed from 2.31 × 10^10^ Pa to 16.09 × 10^10^ Pa, from 0.54 × 10^7^ Pa to 2.59 × 10^7^ Pa, from 1.66 × 10^6^ Pa to 10.11 × 10^6^ Pa, from −4.98 × 10^−7^ K^−1^ to −5.89 × 10^−7^ K^−1^, and from 5.72 × 10^−4^ K^−1^ to 1.66 × 10^−4^ K^−1^, respectively. The mechanism by which Vf_c_ influences the properties of carbon-fiber-reinforced rubber was revealed through the distribution of Von Mises stress. This research will contribute to improving the performance of carbon-fiber-reinforced rubber and promote its application.

## 1. Introduction

Rubber products have been widely applied in many fields [1], and their properties are improved by the addition of certain fillers [2,3,4,5,6,7,8,9,10]. Alam et al. [2] improved the mechanical and energy-harvesting abilities of low-cost natural rubber composites made of stearic-acid-modified diatomaceous earth (mDE) and carbon nanotubes (CNTs). The thermal and mechanical performance of carbon-based rubber nanocomposites was reviewed by Shahamatifard et al. [4], whose assessment included the different fillers of carbon black, carbon nanotubes, and graphene derivatives. Sekar et al. [6] selected biobased hydrothermally treated (HTT) lignin as a potential functional filler for a solution styrene butadiene and butadiene rubber blend, which could achieve the complex reinforcement phenomena of using a silane as coupling agent for hydrothermally treated lignin. Wang et al. [8] proved that the strength of the filler–rubber interfacial interaction gradually increased with the addition of hydroxyl groups for similar dispersion states, which would assist with the design of more desirable silica structures with a highly branched structure and multiple hydroxyl groups as an ideal reinforcing agent. Tagliaro et al. [10] studied the possible synergistic self-assembly of nanosilica and sepiolite in the generation of a cooperative hybrid filler network in rubber-based nanocomposites, which was able to enhance the properties of the rubber materials. These studies [2,3,4,5,6,7,8,9,10] improve the performance of rubber products and promote their practical applications.

Among functional fillers, advanced fiber materials, such as aramid fibers [11,12,13,14], carbon fibers [15,16,17,18], and polyethylene fibers [19,20,21,22], can significantly improve the properties of rubber products due to their excellent performance. Inspired by mussel adhesion proteins and mimicking adhesive molecules, a new environmentally friendly dipping system was developed by Zhang et al. [11,12], which was able to improve the interfacial adhesion between aramid fiber and rubber. Yin et al. [13,14] investigated the fatigue behaviors of aramid fiber/styrene butadiene rubber filled with carbon black in terms of changes in the fatigue strain; this method effectively suppressed the propagation of fatigue cracks due to the use of fibers with a well-bonded flexible interface layer. Mei et al. [15] utilized carbon fibers, graphene, and carbon nanotubes to prepare a carbon-based nano-conductive silicone rubber via solution blending; it exhibited good properties and great application prospects. Akbolat et al. studied the influences of hybrid toughening—via core–shell rubber particles and non-woven thermoplastic veils—on the delamination resistance, crack migration, and R-curve behaviors in carbon fiber/epoxy laminates [17]; this approach made it harder for potential cracks to grow. To improve interfacial adhesion to the rubber matrix, a lower-cost surface modification strategy with ultrahigh-molecular-weight polyethylene fibers was proposed by Fang et al. [19,20]; they replaced expensive dopamine with the two-component system of catechol/tetraethylenepentamine, followed by deposits of nano zinc oxide. He et al. [21] used ultra-high-molecular-weight polyethylene short fibers with a diameter of 20 μm and a length of 2 cm as rubber fillers, which helped to improve the tear resistance of industrially prepared rubber conveyor belts.

In studying the overall performance of fiber-reinforced rubber, many detection methods have been utilized to characterize its various properties, such as its mechanical properties [23,24,25], thermal properties [26,27,28,29], sound insulation properties [30,31], aging resistance [32,33], and sealing properties [34,35]. Mechanical tests of unidirectional fiber composites and quasi-isotropic short-fiber composites for human-hair-reinforced nitrile butadiene rubbers were performed by Statnik et al. [23] under tension using a miniature universal testing machine. Chao et al. [24] examined the tensile and flexural properties of jute-fiber-reinforced polymers using ASTM D3039 [36] and ASTM D7264 [37] standards, and the influences of the end condition, the thickness ratio of the core to the face layer, and the thickness ratio of the face layer to the core on the dynamic properties of sandwich panels were presented. A carboxyl-terminated butadiene acrylonitrile liquid rubber, toughened with silicon carbide and a stitched E-glass fiber-reinforced epoxy composite, was prepared by Velmurugan et al. [27]; it was further tested for its relative effect on the material’s thermal, wear, visco-elastic, and fatigue behaviors. Mahendra et al. [29] estimated the reinforcing effect of nanocrystalline cellulose and nanofiber cellulose and its effect as a compatibilizing agent of a polypropylene/cyclic natural rubber blend in terms of its mechanical and thermal properties. El-Wakil et al. [30] showed that a styrene–butadiene rubber composite containing 10 phr of maleates of Eichhornia crassipes fiber had a sound absorption amplitude equal to 0.9 at a frequency of 400 Hz. Miedzianowska et al. [32] researched the effect of different ratios of mixed polymers and the amount of filler on the material’s tensile strength and elongation at break, resistance to thermo-oxidative aging, tear resistance, hardness, barrier and damping properties, and flammability. The mechanical behaviors of a non-orthogonal reticulated fabric-reinforced rubber composite and its complex fabric rubber seal were investigated by Xu et al. [34], and the macroscopic behaviors of the complex fabric rubber seal were studied through experiments and a numerical simulation, which demonstrated the suitability of the proposed macro-mesoscopic characterization approach. The characterizations of the materials’ various abilities provide a scientific basis for the promotion of the practical applications of all kinds of rubber products.

To improve research efficiency and reduce the test costs, finite element simulations have been used to analyze the properties of rubber products [38,39,40,41,42,43,44,45]. Yang and Lou [39] investigated the effects of oxidative aging on the static and dynamic properties of nitrile rubber at the molecular scale using a molecular dynamics simulation, which provided insights into the effect of molecular changes due to oxidative aging on the structural and dynamic properties of rubber materials at the molecular level. Singaravel et al. [41] reviewed the experimental and molecular dynamics simulations of the mechanical properties of carbon-nanotube-reinforced rubber-based composites, presenting the key mechanical properties of these nanocomposites. A molecular dynamics simulation and dissipative particle dynamics simulation were carried out by Ma et al. [43] to predict the compatibility of natural rubber and chloroprene rubber in view of Flory–Huggins parameters. Sodeifian et al. [44] studied the molecular dynamics of epoxy/clay nanocomposites, proving that the chain relaxation process was slowed by polymer–particle interactions. The effect of the impact angle on the resistance of a foamed silicone rubber sandwich structure against low-velocity impacts was studied by Zhang et al. [45] using the finite element method.

In this study, the mechanical and thermal properties of carbon-fiber-reinforced rubber were investigated using finite element simulations; we aimed to study the influence rule and action mechanism. First, a simplified micromechanical model of carbon-fiber-reinforced rubber was constructed using a repeatable unit cell (RUC) with the periodic boundary condition, ensuring high simulation accuracy while improving the simulation efficiency. Second, the mechanical and thermal properties of carbon-fiber-reinforced rubber were investigated in the fiber volume fraction range of 10–70% with the interval of 5%; the investigated properties included the longitudinal Young’s modulus, the transversal Young’s modulus, the in-plane shear modulus, the longitudinal thermal expansion coefficient, and the transversal thermal expansion coefficient. Third, the corresponding theoretical mechanical and thermal properties were derived using various rule of mixture (ROM) models, such as the Voigt–Reuss model [46], the modified Voigt–Reuss model [47], the Chamis model [48], the Halpin–Tsai model [49], and the Halpin–Tsai–Nielsen model [50], which enabled us to conduct a contrastive analysis with the finite simulation results. Fourth, the influencing mechanism of the fiber volume fraction in relation to the properties of the carbon-fiber-reinforced rubber was revealed by examining the distribution of stress with various load types; this offered a visual explanation of the reasons for differences in properties with the various parameters. The abbreviations used in this study and their corresponding full titles are summarized in the Table A1 (Appendix A). This research method is not only applicable to the carbon-fiber-reinforced rubber examined in this study but also can be used to study the properties of other fillers in reinforced rubber products; it can also provide an effective reference for the analysis of other polymer composites.

## 2. Simplified Micromechanical Model of Carbon-Fiber-Reinforced Rubber

A simplified micromechanical model for carbon-fiber-reinforced rubber was built based on the basic theory of solid mechanics, as shown in Figure 1, which was the foundation for our study of the material’s mechanical and thermal properties. In the geometric model in Figure 1a, the central pink cylinder represents the carbon fiber, and the surrounding green region represents the rubber, which could expand in each direction with the periodic boundary condition. The length of the side for this cubic unit cell was set as 1, and the volume fraction of the carbon fiber Vf_c_ and the corresponding fraction of rubber Vf_r_ (Vf_r_ = 1 − Vf_c_) could be adjusted by changing the diameter of the carbon fiber d_c_. Taking into consideration normal carbon-fiber-reinforced rubber, the research scope for Vf_c_ was set to 10% to 70% with the interval of 5%; it was selected as the independent variable in this research. Afterward, the model was gridded and meshed, as shown in Figure 1b. The maximum cell size, minimum cell size, maximum cell-growth rate, curvature factor, and narrow region resolution were set to 2 mm, 0.02 mm, 1.3, 0.2, and 1, respectively. A free triangular mesh was utilized in this research, and the whole domain was constructed by the swept part. In terms of the cell periodicity for the mechanical properties, the boundary condition was set as the mean strain and the computed average attribute was set to the standard elastic matrix for each boundary pair. Homoplastically, in terms of the cell periodicity for thermal properties, the boundary condition was set as free expansion, and the computed average attribute was selected as the thermal expansion coefficient for each boundary pair. Using these methods, the stress envelope of the RUC was obtained, as shown in Figure 1c.

The carbon fiber was treated as an orthotropic material, and the rubber was considered to be an isotropic material. Thus, the selected parameters for the carbon fiber and for the rubber are summarized in Table 1 and Table 2, respectively. The symbols of E_xc_, E_yc_, and E_zc_ represent the Young’s modulus of the carbon fiber within the x-axis direction, the y-axis direction, and the z-axis direction, respectively, while G_xyc_, G_yzc_, and G_xzc_ represent the shear modulus in the x–y oblique plane, the y–z oblique plane, and the x–z oblique plane, respectively. Accordingly, the symbols of υ_xyc_, υ_yzc_, and υ_xzc_ represent the Poisson’s ratio in the x-axis direction, the y-axis direction, and the z-axis direction, respectively, while α_xyc_, α_yzc_, and α_xzc_ represent the thermal expansion coefficient in the x-axis direction, y-axis direction, and z-axis direction, respectively. The Young’s modulus of the carbon fiber was significantly larger than that of the rubber. Therefore, the carbon fiber was able to reinforce the mechanical and thermal properties of the base material. Meanwhile, various kinds of loads for the cell periodicity were investigated in this study, as shown in Table 3. According to the selected parameters shown in Table 1, Table 2, and Table 3, the longitudinal Young’s modulus, transversal Young’s modulus, in-plane shear modulus, longitudinal thermal expansion coefficient, and transversal thermal expansion coefficient could be obtained using the simplified micromechanical model shown in Figure 1 based on the solid mechanics theory.

## 3. Theoretical ROM Models for Carbon-Fiber-Reinforced Rubber

The theoretical mechanical and thermal properties for carbon-fiber-reinforced rubber can be derived using various ROM models. Thus, the Voigt–Reuss model [46], modified Voigt–Reuss model [47], Chamis model [48], Halpin–Tsai model [49], and Halpin–Tsai–Nielsen model [50] were used in this study; these can be considered a mode of analysis that contrasts with the simulation results.

### 3.1. Young’s Modulus

The Young’s modulus E can be derived using the Voigt–Reuss model [46], modified Voigt–Reuss model [47], Chamis model [48], Halpin–Tsai model [49], and Halpin–Tsai–Nielsen model [50]. The computational formulas are as follows.

According the Voigt–Reuss model [46], the Young’s modulus E_xx−VR_, E_yy−VR_, and E_zz−VR_ can be obtained using Equations (1)–(3), respectively. We observed that E_yy−VR_ = E_zz−VR_, because E_yc_ was equal to E_zc_ according to the parameters presented in Table 1:(1)$$E_{xx-VR}=Vf_{c}\cdot E_{xc}+Vf_{r}\cdot E_{r}$$
(2)$$E_{yy-VR}=\frac{1}{Vf_{c}/E_{yc}+Vf_{r}/E_{r}}=\frac{E_{r}\cdot E_{yc}}{Vf_{c}\cdot E_{r}+Vf_{r}\cdot E_{yc}}$$
(3)$$E_{zz-VR}=\frac{1}{Vf_{c}/E_{zc}+Vf_{r}/E_{r}}=\frac{E_{r}\cdot E_{zc}}{Vf_{c}\cdot E_{r}+Vf_{r}\cdot E_{zc}}$$

According to the modified Voigt–Reuss model [47], the Young’s modulus E_xx−MVR_, E_yy−MVR_, and E_zz−MVR_ can be obtained using Equations (4)–(6), respectively. Here, η_c_ and η_r_ were the correction factors for the carbon fiber and the rubber matrix and can be derived from Equations (7) and (8), respectively. The meanings of these symbols are the same as in Table 1 and Table 2:(4)$$E_{xx-MVR}=Vf_{c}\cdot E_{xc}+Vf_{r}\cdot E_{r}$$
(5)$$E_{yy-MVR}=\frac{1}{\eta_{c}Vf_{c}/E_{yc}+\eta_{r}Vf_{r}/E_{r}}=\frac{E_{r}\cdot E_{yc}}{\eta_{c}\cdot Vf_{c}\cdot E_{r}+\eta_{r}\cdot Vf_{r}\cdot E_{yc}}$$
(6)$$E_{zz-MVR}=\frac{1}{\eta_{c}Vf_{c}/E_{zc}+\eta_{r}Vf_{r}/E_{r}}=\frac{E_{r}\cdot E_{zc}}{\eta_{c}\cdot Vf_{c}\cdot E_{r}+\eta_{r}\cdot Vf_{r}\cdot E_{zc}}$$
(7)$$\eta_{c}=\frac{Vf_{c}\cdot E_{yc}+((1-\upsilon_{xyc}^{2}\cdot \frac{E_{yc}}{E_{xc}})\cdot E_{r}+\upsilon_{r}\cdot \upsilon_{xyc}\cdot E_{yc})\cdot Vf_{r}}{E_{xx-MVR}}$$
(8)$$\eta_{r}=\frac{Vf_{r}\cdot E_{r}+((1-\upsilon_{r}^{2})\cdot E_{c}+\upsilon_{r}\cdot \upsilon_{xyc}\cdot E_{r})\cdot Vf_{c}}{E_{xx-MVR}}$$

Similarly, according to the Chamis model [48], the Young’s modulus E_xx−C_, E_yy−C_, and E_zz−C_ values can be obtained using Equations (9)–(11), respectively:(9)$$E_{xx-C}=Vf_{c}\cdot E_{xc}+Vf_{r}\cdot E_{r}$$
(10)$$E_{yy-C}=\frac{E_{r}}{1-\sqrt{Vf_{c}}\cdot \left(1-E_{r}/E_{yc}\right)}$$
(11)$$E_{yy-C}=\frac{E_{r}}{1-\sqrt{Vf_{c}}\cdot \left(1-E_{r}/E_{zc}\right)}$$

Analogously, according to the Halpin–Tsai model [49], the Young’s modulus E_xx−HT_, E_yy−HT_, and E_zz−HT_ values can be obtained using Equations (12)–(14), respectively. Here, η_xc_, η_yc_, and η_zc_ were the correction factors in the x-axis direction, y-axis direction, and z-axis direction and are obtained using Equations (15)–(17), respectively. ζ_xc_, ζ_yc_, and ζ_zc_ were the enhancement factors in the x-axis direction, y-axis direction, and z-axis direction, respectively, and their values were set to 2 for normal conditions in this study:(12)$$E_{xx-HT}=\frac{1+\zeta_{xc}\cdot \eta_{xc}\cdot Vf_{c}}{1-\eta_{xc}\cdot Vf_{c}}\cdot E_{r}$$
(13)$$E_{yy-HT}=\frac{1+\zeta_{yc}\cdot \eta_{yc}\cdot Vf_{c}}{1-\eta_{yc}\cdot Vf_{c}}\cdot E_{r}$$
(14)$$E_{zz-HT}=\frac{1+\zeta_{zc}\cdot \eta_{zc}\cdot Vf_{c}}{1-\eta_{zc}\cdot Vf_{c}}\cdot E_{r}$$
(15)$$\eta_{xc}=\frac{E_{xc}/E_{r}-1}{E_{xc}/E_{r}+\zeta_{xc}}$$
(16)$$\eta_{yc}=\frac{E_{yc}/E_{r}-1}{E_{yc}/E_{r}+\zeta_{yc}}$$
(17)$$\eta_{zc}=\frac{E_{zc}/E_{r}-1}{E_{zc}/E_{r}+\zeta_{zc}}$$

Similarly, according to the Halpin–Tsai–Nielsen model [50], the Young’s modulus E_xx−HTN_, E_yy−HTN_, and E_zz−HTN_ values can be obtained using Equations (18)–(20), respectively. Here, the values of η_xc_, η_yc_, η_zc_, ζ_xc_, ζ_yc_, and ζ_zc_ were the same as those in the Halpin–Tsai model [49]. Ψ was the packing factor, which could be calculated using Equation (21). ϕ_max_ was the maximum filling rate, and here it was set to 0.8 for normal conditions:(18)$$E_{xx-HTN}=\frac{1+\zeta_{xc}\cdot \eta_{xc}\cdot Vf_{c}}{1-\eta_{xc}\cdot \psi \cdot Vf_{c}}\cdot E_{r}$$
(19)$$E_{yy-HTN}=\frac{1+\zeta_{yc}\cdot \eta_{yc}\cdot Vf_{c}}{1-\eta_{yc}\cdot \psi \cdot Vf_{c}}\cdot E_{r}$$
(20)$$E_{zz-HTN}=\frac{1+\zeta_{zc}\cdot \eta_{zc}\cdot Vf_{c}}{1-\eta_{zc}\cdot \psi \cdot Vf_{c}}\cdot E_{r}$$
(21)$$\psi =\left(\frac{1-\phi_{\max}}{\phi_{\max}^{2}}\right)\cdot Vf_{c}$$

### 3.2. Shear Modulus

The shear modulus G can be derived using the Voigt–Reuss model [46], the modified Voigt–Reuss model [47], the Chamis model [48], the Halpin–Tsai model [49], and the Halpin–Tsai–Nielsen model [50]. The computational formulas are as follows.

According the Voigt–Reuss model [46], the shear modulus G_xy−VR_, G_yz−VR_, and G_xz−VR_ values can be obtained using Equations (22)–(24), respectively. We also found that G_xy−VR_ = G_xz−VR_, because G_xyc_ is equal to E_xzc_ according to the parameters shown in Table 1:(22)$$G_{xy-VR}=\frac{1}{Vf_{c}/G_{xyc}+Vf_{r}/G_{r}}$$
(23)$$G_{yz-VR}=\frac{1}{Vf_{c}/G_{yzc}+Vf_{r}/G_{r}}$$
(24)$$G_{xz-VR}=\frac{1}{Vf_{c}/G_{xzc}+Vf_{r}/G_{r}}$$

According to the modified Voigt–Reuss model [47], the shear modulus G_xy−MVR_, G_yz−MVR_, and G_xz−MVR_ values can be obtained using Equations (25)–(27), respectively. Here, η_g_ was the correction factor, which was set to 0.6 for normal conditions:(25)$$G_{xy-MVR}=\frac{\left(Vf_{c}+\eta_{g}\cdot Vf_{r}\right)\cdot G_{xyc}\cdot G_{r}}{Vf_{c}\cdot G_{r}+Vf_{r}\cdot G_{xyc}}$$
(26)$$G_{yz-MVR}=\frac{1}{Vf_{c}/G_{yzc}+Vf_{r}/G_{r}}$$
(27)$$G_{xz-MVR}=\frac{\left(Vf_{c}+\eta_{g}\cdot Vf_{r}\right)\cdot G_{xzc}\cdot G_{r}}{Vf_{c}\cdot G_{r}+Vf_{r}\cdot G_{xzc}}$$

According to the Chamis model [48], the shear modulus values for G_xy−C_, G_yz−C_, and G_xz−C_ can be obtained using Equations (28)–(30), respectively:(28)$$G_{xy-C}=\frac{G_{r}}{1-\sqrt{Vf_{c}}\cdot \left(1-G_{r}/G_{xyc}\right)}$$
(29)$$G_{yz-C}=\frac{G_{r}}{1-\sqrt{Vf_{c}}\cdot \left(1-G_{r}/G_{yzc}\right)}$$
(30)$$G_{xz-C}=\frac{G_{r}}{1-\sqrt{Vf_{c}}\cdot \left(1-G_{r}/G_{xzc}\right)}$$

Similarly, according to the Halpin–Tsai model [49], the shear modulus values for G_xy−HT_, G_yz−HT_, and G_xz−HT_ can be determined using Equations (31)–(33), respectively. Here, η_xyc_, η_yzc_, and η_xzc_ were the correction factors in the x–y oblique plane, the y–z oblique plane, and the x–z oblique plane, which can be obtained using Equations (34)–(36), respectively. ζ_xyc_, ζ_yzc_, and ζ_xzc_ were the enhancement factors in the x–y oblique plane, y–z oblique plane, and x–z oblique plane, respectively, and their values were set to 1 for normal conditions in this research:(31)$$G_{xy-HT}=\frac{1+\zeta_{xyc}\cdot \eta_{xyc}\cdot Vf_{c}}{1-\eta_{xyc}\cdot Vf_{c}}\cdot G_{r}$$
(32)$$G_{yz-HT}=\frac{1+\zeta_{yzc}\cdot \eta_{yzc}\cdot Vf_{c}}{1-\eta_{yzc}\cdot Vf_{c}}\cdot G_{r}$$
(33)$$G_{xz-HT}=\frac{1+\zeta_{xzc}\cdot \eta_{xzc}\cdot Vf_{c}}{1-\eta_{xzc}\cdot Vf_{c}}\cdot G_{r}$$
(34)$$\eta_{xyc}=\frac{G_{xyc}/G_{r}-1}{G_{xyc}/G_{r}+\zeta_{xyc}}$$
(35)$$\eta_{yzc}=\frac{G_{yzc}/G_{r}-1}{G_{yzc}/G_{r}+\zeta_{yzc}}$$
(36)$$\eta_{xzc}=\frac{G_{xzc}/G_{r}-1}{G_{xzc}/G_{r}+\zeta_{xzc}}$$

Similarly, according to the Halpin–Tsai–Nielsen model [50], the shear modulus G_x−HTN_, G_yz−HTN_, and G_xz−HTN_ values can be obtained using Equations (37)–(39), respectively. Here, η_xyc_, η_yzc_, η_xzc_, ζ_xyc_, ζ_yzc_, and ζ_xzc_ are the same as in Equations (34)–(36). Ψ was the packing factor, which was the same as in Equation (20), and it can also be calculated using Equation (21):(37)$$G_{xy-HTN}=\frac{1+\zeta_{xyc}\cdot \eta_{xyc}\cdot Vf_{c}}{1-\eta_{xyc}\cdot \psi \cdot Vf_{c}}\cdot G_{r}$$
(38)$$G_{yz-HTN}=\frac{1+\zeta_{yzc}\cdot \eta_{yzc}\cdot Vf_{c}}{1-\eta_{yzc}\cdot \psi \cdot Vf_{c}}\cdot G_{r}$$
(39)$$G_{xz-HTN}=\frac{1+\zeta_{xzc}\cdot \eta_{xzc}\cdot Vf_{c}}{1-\eta_{xzc}\cdot \psi \cdot Vf_{c}}\cdot G_{r}$$

### 3.3. Thermal Expansion Coefficient

The thermal expansion coefficient α can be calculated using the Voigt–Reuss model [46] and the Chamis model [48]. According the Voigt–Reuss model [46], the thermal expansion coefficient α_xx−VR_, α_yy−VR_, and α_zz−VR_ values are given by Equations (40)–(42), respectively. We found that α_yy−VR_ = α_zz−VR_, because α_yc_ is equal to α_zc_ according to the parameters given in Table 1:(40)$$\alpha_{xx-VR}=\frac{Vf_{c}\cdot \alpha_{xc}\cdot E_{xc}+Vf_{r}\cdot \alpha_{r}\cdot E_{r}}{Vf_{c}\cdot E_{xc}+Vf_{r}\cdot E_{r}}$$
(41)$$\alpha_{yy-VR}=Vf_{c}\cdot \alpha_{yc}+Vf_{r}\cdot \alpha_{r}$$
(42)$$\alpha_{zz-VR}=Vf_{c}\cdot \alpha_{zc}+Vf_{r}\cdot \alpha_{r}$$

Similarly, according to the Chamis model [48], the thermal expansion coefficient α_xx−C_, α_yy−C_, and α_zz−C_ values can be obtained using Equations (43)–(45), respectively:(43)$$\alpha_{xx-C}=\frac{Vf_{c}\cdot \alpha_{xc}\cdot E_{xc}+Vf_{r}\cdot \alpha_{r}\cdot E_{r}}{Vf_{c}\cdot E_{xc}+Vf_{r}\cdot E_{r}}$$
(44)$$\alpha_{yy-C}=\sqrt{Vf_{c}}\cdot \alpha_{yc}+\left(1-\sqrt{Vf_{c}}\right)\cdot \alpha_{r}\cdot \left(1+\frac{Vf_{c}\cdot \upsilon_{r}\cdot E_{xc}}{Vf_{c}\cdot E_{xc}+Vf_{r}\cdot E_{r}}\right)$$
(45)$$\alpha_{zz-C}=\sqrt{Vf_{c}}\cdot \alpha_{zc}+\left(1-\sqrt{Vf_{c}}\right)\cdot \alpha_{r}\cdot \left(1+\frac{Vf_{c}\cdot \upsilon_{r}\cdot E_{xc}}{Vf_{c}\cdot E_{xc}+Vf_{r}\cdot E_{r}}\right)$$

### 3.4. Poisson’s Ratio

According to the Voigt–Reuss model [46], the modified Voigt–Reuss model [47], the Halpin–Tsai model [49], and the Halpin–Tsai–Nielsen model [50], the calculation formulas for the Poisson’s ratio υ_xy−VR_, υ_yz−VR_, and υ_xz−VR_ values are the same. They are given by Equations (46)–(48), respectively. In addition, for the Chamis model [48], the Poisson’s ratio υ_xy−VR_ and υ_xz−VR_ values can be obtained using Equations (46) and (48), respectively, and the υ_yz−VR_ value can be calculated according to Equation (49):(46)$$\upsilon_{xy}=Vf_{c}\cdot \alpha_{xyc}+Vf_{r}\cdot \alpha_{r}$$
(47)$$\upsilon_{yz}=Vf_{c}\cdot \alpha_{yzc}+Vf_{r}\cdot \alpha_{r}$$
(48)$$\upsilon_{xz}=Vf_{c}\cdot \alpha_{xzc}+Vf_{r}\cdot \alpha_{r}$$
(49)$$\upsilon_{yz-C}=\frac{E_{yy-C}}{2G_{yz-C}}-1$$

## 4. Results and Discussion

Equations (1)–(49) in the theoretical model show that some of these pairs of performance parameters were the same, because the rubber matrix is a homogeneous material and the carbon fiber has similar properties in the y-axis direction and the z-axis direction. Thus, taking the actual applications into account, the longitudinal Young’s modulus E_xx_, transversal Young’s modulus E_yy_, in-plane shear modulus G_xy_, longitudinal thermal expansion coefficient α_xx_, and transversal thermal expansion coefficient α_yy_ were selected as the representative parameters to compare the simulation results with the theoretical data obtained using the various ROM models. This could be considered the mutual verification of the finite element simulation model and the theoretical model.

### 4.1. Longitudinal Young’s Modulus E_xx_

The comparisons between the simulation results and the theoretical data of the longitudinal Young’s modulus E_xx_ are shown in Figure 2. It is clear that the longitudinal Young’s modulus E_xx_ value obtained using the finite element simulation was completely consistent with the values determined using the various ROM models. The main reason for this phenomenon is that the Young’s modulus of the carbon fiber in the x-axis direction E_xc_ (230 GPa) was far larger than that of the rubber matrix E_r_ (4 MPa), meaning that the carbon fiber took most of the load. Meanwhile, the binding force between the carbon fiber and the rubber matrix was large in the x-axis direction, which indicated that the longitudinal Young’s modulus E_xx_ was slightly smaller than that of carbon fiber itself. Furthermore, Equations (1), (4), (9), (12), and (18) suggest that there were almost no differences between the various ROM models when calculating the longitudinal Young’s modulus E_xx_.

### 4.2. Transversal Young’s Modulus E_yy_

The comparison between the simulation results and the theoretical data for the transversal Young’s modulus E_yy_ are shown in Figure 3. The simulation results tended to be consistent with those of the theoretical data obtained using the various ROM models, and the deviations among the various ROM models can be determined using Equations (2), (5), (10), (13), and (19). Meanwhile, the E_yy_ in Figure 3 was obviously smaller than the E_xx_ in Figure 2, because the binding force between the carbon fiber and the rubber matrix was small in the y-axis direction and the E_yy_ value was slightly larger than E_r_.

### 4.3. In-Plane Shear Modulus G_xy_

The comparison between the simulation results and the theoretical data for the in-plane shear modulus G_xy_ are shown in Figure 4. The theoretical data obtained using the various ROM models were derived using Equations (22), (25), (28), (31), and (37). The simulation results were consistent with the theoretical data. Meanwhile, the in-plane shear modulus G_xy_ ranged from 2 MPa to 10 MPa as the Vfc increased from 10% to 70%, which was slightly larger than the shear modulus of rubber matrix G_r_ (1.36 GPa); this is because the binding force between the carbon fiber and the rubber matrix was small in the x–y oblique plane. Although the shear modulus of the carbon fiber in x–y oblique plane G_xyc_ was large, the brittle binding force between the matrix and the reinforcement made the composite material more susceptible to damage. 

### 4.4. Longitudinal Thermal Expansion Coefficient α_xx_

The comparison between the simulation results and the theoretical data for the longitudinal thermal expansion coefficient α_xx_ are shown in Figure 5. The α_xx_ had a negative value, and it decreased as the Vfc increased from 10% to 70%; this is because the longitudinal thermal expansion coefficient of the carbon fiber α_xc_ was −0.8 × 10^−6^ K^−1^. The theoretical data agreed well with the simulation results, which could be considered to constitute mutual verification. The theoretical data obtained using the Voigt–Reuss model were identical to those obtained using the Chamis model, because their computational formulas were same; this is indicated by Equations (40) and (43).

### 4.5. Transversal Thermal Expansion Coefficient α_yy_

The comparison between the simulation results and the theoretical data for the transversal thermal expansion coefficient α_yy_ are shown in Figure 6. Unlike the corresponding longitudinal thermal expansion coefficient α_xx_, shown in Figure 5, the α_yy_ value was positive and fell between the thermal expansion coefficient of the carbon fiber α_yc_ (25 × 10^−6^ K^−1^) and that of the rubber matrix α_r_ (1 × 10^−4^ K^−1^). Meanwhile, we found slight deviations between the theoretical data obtained using the Voigt–Reuss model and those produced by the Chamis model, which can be assessed according to their computational formulas in Equations (41) and (44). Furthermore, the consistency between the simulation results and the theoretical data can be considered to verify their mutual reliability.

Taking the finite element simulation result as the reference value, the relative deviations for the various ROM models can be derived from Equation (50), and the results are shown in Figure 7. Here, D_r_, data_ROM_, and data_simulation_ represent the relative deviation, the theoretical data for each ROM model, and the simulation result, respectively:(50)$$D_{r}=\frac{data_{ROM}-data_{simulation}}{data_{simulation}}$$

Figure 7 indicates that, for different properties, the deviations for the various ROM models were not the same. For the longitudinal Young’s modulus E_xx_, the deviation for each ROM model was almost the same, as is consistent with the results shown in Figure 2. Similarly, for the longitudinal thermal expansion coefficient α_xx_, the deviation for the Voigt–Reuss model was almost the same as that for the Chamis model. However, for the transversal Young’s modulus E_yy_, the in-plane shear modulus G_xy_, and the transversal thermal expansion coefficient α_yy_, the corresponding ROM models with the minimum deviation were the modified Voigt–Reuss model, the Halpin–Tsai model, and the Voigt–Reuss model, respectively. Furthermore, we found that the deviation was also affected by the volume fraction of the carbon fiber Vf_c_. Taking the relative deviation for the transversal Young’s modulus E_yy_ in Figure 7b, for example, the modified Voigt–Reuss model obtained higher prediction accuracy when the Vf_c_ was smaller than 0.6, and the thermotical data obtained using the Halpin–Tsai–Nielsen model were closer to the simulation results when the Vf_c_ was larger than 0.6. Analogously, regarding the deviation for the in-plane shear modulus G_xy_ in Figure 7c, the Halpin–Tsai model achieved higher prediction accuracy with a smaller Vf_c_ value, and the Chamis model achieved better accuracy with a larger Vf_c_. Thus, each ROM model is suitable for a particular scenario, and the appropriate ROM model should be selected to study the various properties of carbon-fiber-reinforced rubber. 

Table 4 shows the comparisons of the longitudinal Young’s modulus E_xx_, transversal Young’s modulus E_yy_, in-plane shear modulus G_xy_, longitudinal thermal expansion coefficient α_xx_, and transversal thermal expansion coefficient α_yy_ between the carbon fiber, the rubber matrix, and the composite material. Here, Vf_c_ = 0% refers to a pure rubber matrix, and Vf_c_ = 100% refers to pure carbon fiber. Meanwhile, the data for the composite material were represented by the finite element simulation value, and those for the carbon fiber and rubber matrix were obtained from Table 1 and Table 2, respectively. We found that the reinforcement of the composite material with carbon fiber increased along with the increase in the Vf_c_ value, which proved the effectiveness of the carbon fiber filler.

## 5. Mechanism Analysis

The mechanism of the composite material comprising the rubber matrix and the carbon fiber was intuitively revealed by the distribution of Von Mises stress (N/m^2^). The anisotropic properties were analyzed according to the condition of different kinds of loads, as shown in Table 3; the mechanical properties and the thermal properties will now be discussed.

### 5.1. Mechanical Properties

#### 5.1.1. Load 11

The distribution of Von Mises stress (N/m^2^) with various fiber volume fractions under the conditions of the load in the x-axis direction is shown in Figure 8. We found that most of the load was placed on the carbon fiber, and the rubber matrix withstood the minimal load; these findings are consistent with the previous analysis of the longitudinal Young’s modulus Exx. The contact area and effective binding force between the carbon fiber and the rubber matrix were large in the x-axis direction. With the increase in the volume fraction of the carbon fiber Vf_c_ from 10% to 70%, the reinforcement of carbon fiber with high strength to the rubber matrix with low strength became more and more significant, which indicated that the composite material could bear a larger load in the x-axis direction.

#### 5.1.2. Load 22

The distribution of Von Mises stress (N/m^2^) with various fiber volume fractions under the conditions of the load in the y-axis direction is shown in Figure 9. Compared with the results shown in Figure 8, the reinforcement of the rubber matrix with carbon fibers in the y-axis direction was far less pronounced than in the x-axis direction. The main reason for this phenomenon is that the binding force between the carbon fiber and the rubber matrix was small in the y-axis direction, since the contact area in the y-axis direction between the two was too small relative to that in the x-axis direction. The results indicate that the bearing capacity of carbon-fiber-reinforced rubber was limited; otherwise, the composite material may be destroyed in the binding region.

#### 5.1.3. Load 33

The distribution of Von Mises stress (N/m^2^) with various fiber volume fractions under the condition of the load in the z-axis direction is shown in Figure 10. The distributions of Von Mises stress shown in Figure 10 were similar to those shown in Figure 9 (rotating 90° in the y–z plane along the x-axis), because the rubber matrix was isotropous and the carbon fiber had the same Young’s modulus in the y-axis direction and the z-axis direction, as can be ascertained from the values of Eyc and Ezc in Table 1. Therefore, the carbon-fiber-reinforced rubber could not withstand a large load in the z-axis direction, for reasons similar to those described for the y-axis direction.

#### 5.1.4. Load 12

Figure 11 shows the distribution of Von Mises stress (N/m^2^) with various fiber volume fractions under the conditions of the load in the x–y oblique plane. The load can be divided into the following categories: along the x-axis and along the y-axis. This indicates that the distribution of Von Mises stress shown in Figure 11 is the result of a coupling effect of the Von Mises stress in Figure 8 and that in Figure 9. However, the value of the Von Mises stress in Figure 11 was closer to that in Figure 9 and far smaller than that in Figure 8, because the weak binding force diminished the reinforcement effect. The main difference between the distributions of Von Mises stress in Figure 11 and in Figure 9 was that the interfaces between the carbon fiber and the rubber matrix bore the main stress when the Vf_c_ was large, as shown by Figure 11f,g. On the other hand, when the load was in the y-axis direction, the main stress was concentrated on the carbon fiber, and the interfaces assumed the secondary stress.

#### 5.1.5. Load 23

Figure 12 shows the distribution of Von Mises stress (N/m^2^) with various fiber volume fractions under the conditions of the load in the y–z oblique plane. Similarly, the load comprised one part along the y-axis and the other along the z-axis, which means that the distribution of Von Mises stress in Figure 12 constituted a coupling effect of the Von Mises stress in Figure 9 and that in Figure 10. Notably, when the volume fraction Vf_c_ was lower than 40%, as shown in Figure 12a–d, the Von Mises stress was mainly distributed in the carbon fiber and the four corners of the rubber matrix. When the Vf_c_ increased from 50% to 70%, the distribution of the Von Mises stress gradually transferred to the interfaces between the carbon fiber and the rubber matrix. In particular, when the Vf_c_ was 70%, the Von Mises stress was mainly distributed in the surface of the carbon fiber and the interfaces between the carbon fiber and the rubber matrix. The material broke most easily in the place where the stress was concentrated, which is a finding that is consistent with the previous analysis.

#### 5.1.6. Load 13

Figure 13 shows the distribution of Von Mises stress (N/m^2^) with various fiber volume fractions under the conditions of the load in the x–z oblique plane. Homoplastically, the load comprises one part along the x-axis and the other along the z-axis, meaning that the distribution of Von Mises stress shown in Figure 13 comprised a coupling effect of the Von Mises stresses in Figure 8 and in Figure 10. The distributions of Von Mises stress under the conditions of load in the x–z oblique plane, as shown in Figure 13, were similar to those presented in Figure 11 (rotating 90° in the y–z plane along the x-axis), because the rubber matrix was isotropous, and the carbon fiber had the same Young’s modulus values in the y-axis direction and the z-axis direction.

### 5.2. Thermal Properties

Figure 14 shows the distribution of Von Mises stress (N/m^2^) with various fiber volume fractions for the thermal properties. Unlike the boundary condition of average strain and the calculation target of the elasticity matrix used for the mechanical properties, the boundary condition of free expansion and the calculation target of the thermal expansion coefficient were selected for the examination of the thermal properties. In the free expansion mode, the Von Mises stress was markedly smaller than in the average strain model. In the x-axis direction, the strain was mainly concentrated on the carbon fiber; this finding was similar to that produced by the analysis of the Von Mises stress with Load 11 in Section 5.1.1. This is why the thermal expansion coefficient of the composite material α_xx_ was close to that of the carbon fiber α_xc_. Homoplastically, in the y-axis direction, the strain was majorly concentrated on the rubber matrix; this was similar to the Von Mises stress with Load 22, discussed in Section 5.1.2, and this is why the thermal expansion coefficient of the composite material α_yy_ was close to that of the rubber matrix α_r_. The thermal expansion coefficient of the composite material α_zz_ in the z-axis direction was similar to the α_yy_ value, as is also indicated in Figure 14.

## 6. Conclusions

Based on the solid mechanics method in a finite element simulation, the mechanical and thermal properties of carbon-fiber-reinforced rubber with a volume fraction ranging from 10 to 70% were investigated. We aimed to identify the influencing law and mechanism, and mutual verifications were conducted using various ROM models. The major achievements of this study are as follows.

(1)According to the mutual verification of the finite element simulation results obtained using a simplified micromechanical model and the theoretical data presented by the various ROM models, we provided a simple and effective method for investigating the properties of carbon-fiber-reinforced rubber. Compared to complex and time-consuming experimental tests, the finite element simulation method not only reduced the experimental cost but could also provide a visual representation of the influencing mechanism.(2)The finite element simulation results indicate that, as the fiber volume fraction Vf_c_ increased from 10% to 70%, the mechanical properties of the longitudinal Young’s modulus E_xx_, the transversal Young’s modulus E_yy_, and the in-plane shear modulus G_xy_ changed from 2.31 × 10^10^ Pa to 16.09 × 10^10^ Pa, from 0.54 × 10^7^ Pa to 2.59 × 10^7^ Pa, and from 1.66 × 10^6^ Pa to 10.11 × 10^6^ Pa, respectively. The reinforcing effect of the carbon fiber mainly worked in the x-axis direction because the fiber was installed in this direction. Meanwhile, the reinforcing effect gradually improved as the fiber volume fraction Vf_c_ increased. Therefore, the carbon fiber can be installed along the desired direction, and the multidirectional reinforcement of the rubber matrix can be obtained using two-dimensional or three-dimensional braided carbon fibers.(3)Similarly, the finite element simulation results suggest that, as the fiber volume fraction Vf_c_ increases from 10% to 70%, the longitudinal thermal expansion coefficient α_xx_ and transversal thermal expansion coefficient α_yy_ changed from −4.98 × 10^−7^ K^−1^ to −5.89 × 10^−7^ K^−1^ and from 5.72 × 10^−4^ K^−1^ to 1.66 × 10^−4^ K^−1^, respectively. Moreover, αxx had a negative value, while αyy had a positive value, and both of them decreased as the fiber volume fraction Vf_c_ increased. Thus, the desired thermal properties for the composite material could be obtained by selecting the carbon fiber with the appropriate dimensions and installing it in the right direction.(4)Using finite element simulation, we determined the influencing mechanism of the volume fraction of carbon fiber Vf_c_ to the distribution of Von Mises stress for the carbon-fiber-reinforced rubber; the resultant picture of stress concentration may provide guidance for subsequent research. The interface between the carbon fiber and the rubber matrix was the weak spot, and the binding force should be improved to further enhance the overall performance of the composite material.

This research promotes the understanding and usage of carbon-fiber-reinforced rubber, which could improve the performance of rubber products and facilitate their practical applications. Moreover, this study provides an effective method for studying the properties and mechanisms of rubber materials reinforced with other various fillers, which could be conducive to promoting the further development of the rubber industry.

## Figures and Tables

**Figure 1 polymers-16-02120-f001:**
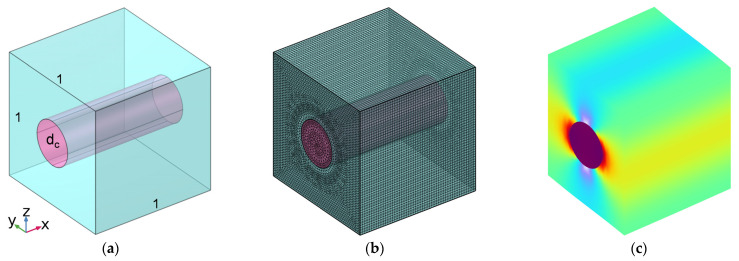
Simplified micromechanical model for carbon-fiber-reinforced rubber. (**a**) Geometric model; (**b**) gridded and meshed model; and (**c**) stress envelope.

**Figure 2 polymers-16-02120-f002:**
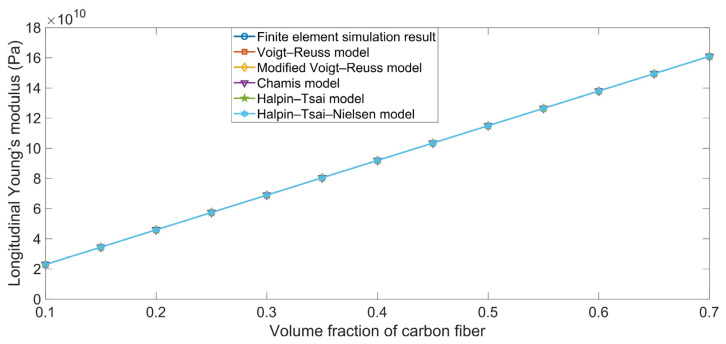
Comparison between the simulation results and the theoretical data obtained using various ROM models of the longitudinal Young’s modulus E_xx_ values.

**Figure 3 polymers-16-02120-f003:**
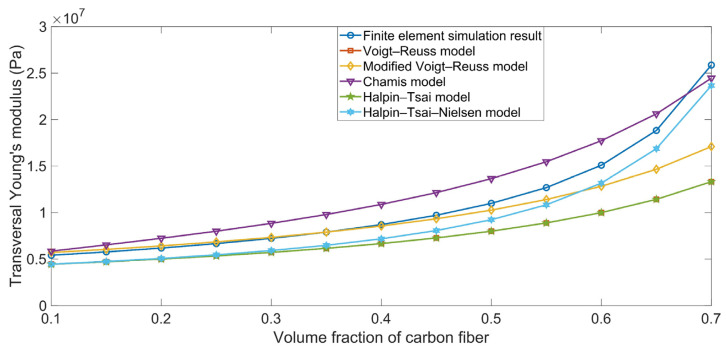
Comparison between the simulation results and theoretical data obtained using various ROM models for the transversal Young’s modulus E_yy_.

**Figure 4 polymers-16-02120-f004:**
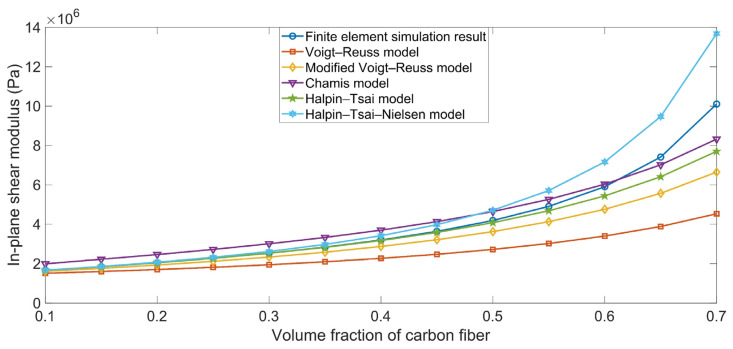
Comparison between simulation results and theoretical data obtained using the various ROM models for the in-plane shear modulus G_xy_.

**Figure 5 polymers-16-02120-f005:**
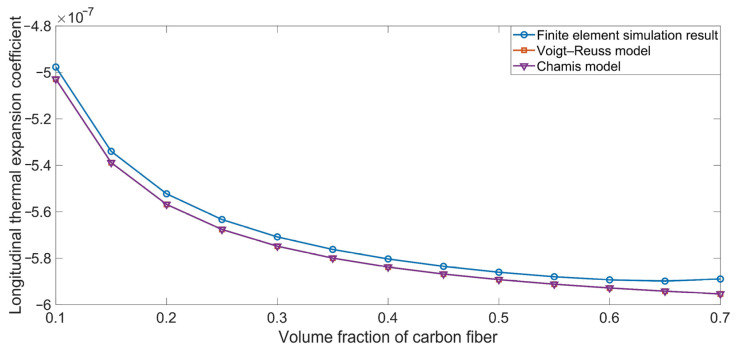
Comparison between the simulation results and the theoretical data obtained using the various ROM models for the longitudinal thermal expansion coefficient α_xx_.

**Figure 6 polymers-16-02120-f006:**
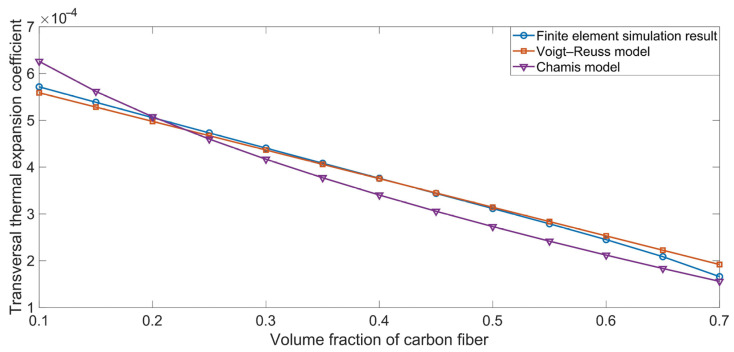
Comparison between the simulation results and theoretical data obtained using various ROM models for the transversal thermal expansion coefficient α_yy_.

**Figure 7 polymers-16-02120-f007:**
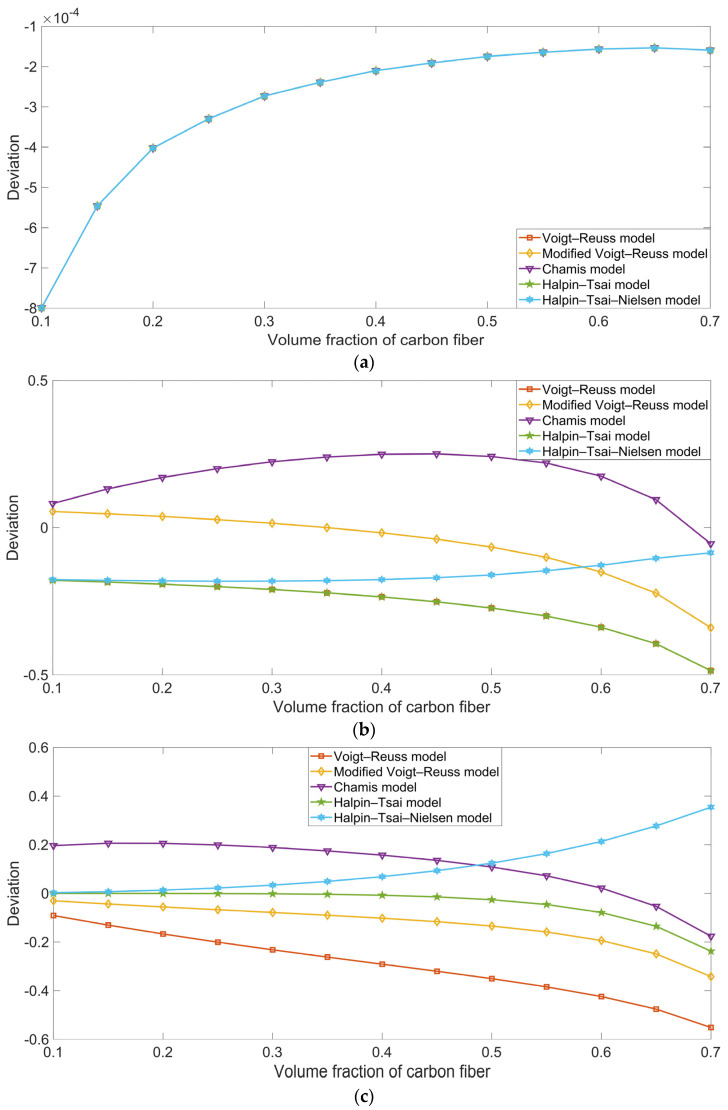
Relative deviations for the various ROM models, taking the finite element simulation result as the reference value. (**a**) Longitudinal Young’s modulus E_xx_; (**b**) transversal Young’s modulus E_yy_; (**c**) in-plane shear modulus G_xy_; (**d**) longitudinal thermal expansion coefficient α_xx_; and (**e**) transversal thermal expansion coefficient α_yy_.

**Figure 8 polymers-16-02120-f008:**
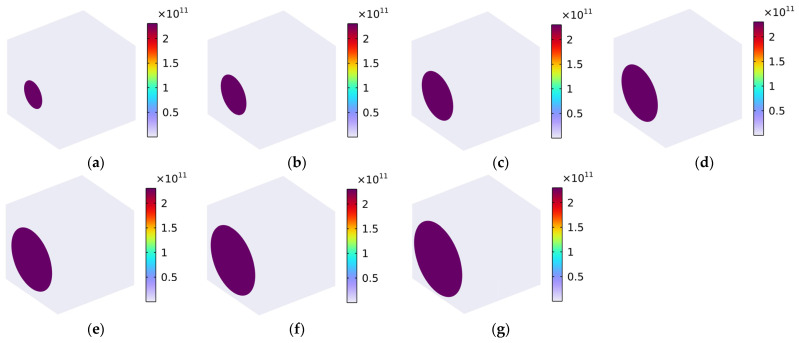
Distribution of Von Mises stress (N/m^2^) with various volume fractions under the conditions of load in the x-axis direction. (**a**) Vf_c_ = 10%; (**b**) Vf_c_ = 20%; (**c**) Vf_c_ = 30%; (**d**) Vf_c_ = 40%; (**e**) Vf_c_ = 50%; (**f**) Vf_c_ = 60%; and (**g**) Vf_c_ = 70%.

**Figure 9 polymers-16-02120-f009:**
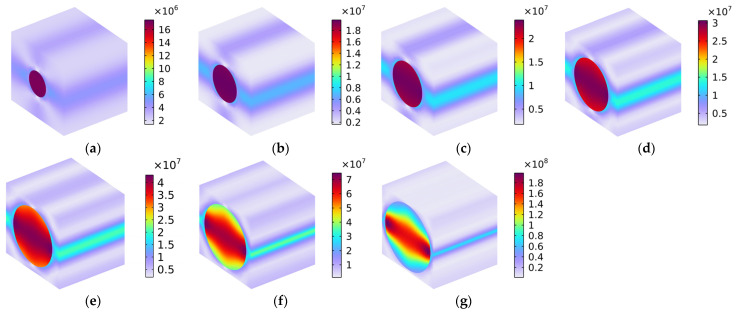
Distribution of Von Mises stress (N/m^2^) with various volume fractions under the condition of load in the y-axis direction. (**a**) Vf_c_ = 10%; (**b**) Vf_c_ = 20%; (**c**) Vf_c_ = 30%; (**d**) Vf_c_ = 40%; (**e**) Vf_c_ = 50%; (**f**) Vf_c_ = 60%; and (**g**) Vf_c_ = 70%.

**Figure 10 polymers-16-02120-f010:**
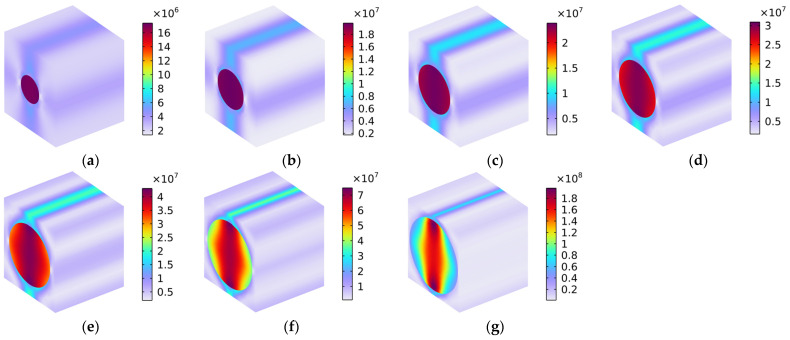
Distribution of Von Mises stress (N/m^2^) with various volume fractions under the conditions of load in the z-axis direction. (**a**) Vf_c_ = 10%; (**b**) Vf_c_ = 20%; (**c**) Vf_c_ = 30%; (**d**) Vf_c_ = 40%; (**e**) Vf_c_ = 50%; (**f**) Vf_c_ = 60%; and (**g**) Vf_c_ = 70%.

**Figure 11 polymers-16-02120-f011:**
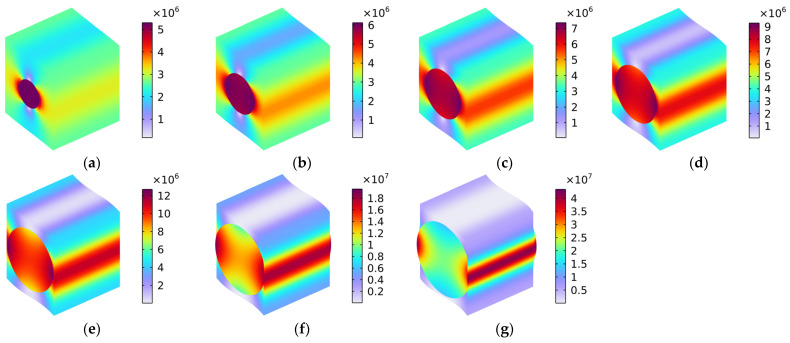
Distribution of Von Mises stress (N/m^2^) with various volume fractions under the conditions of load in the x–y oblique plane. (**a**) Vf_c_ = 10%; (**b**) Vf_c_ = 20%; (**c**) Vf_c_ = 30%; (**d**) Vf_c_ = 40%; (**e**) Vf_c_ = 50%; (**f**) Vf_c_ = 60%; and (**g**) Vf_c_ = 70%.

**Figure 12 polymers-16-02120-f012:**
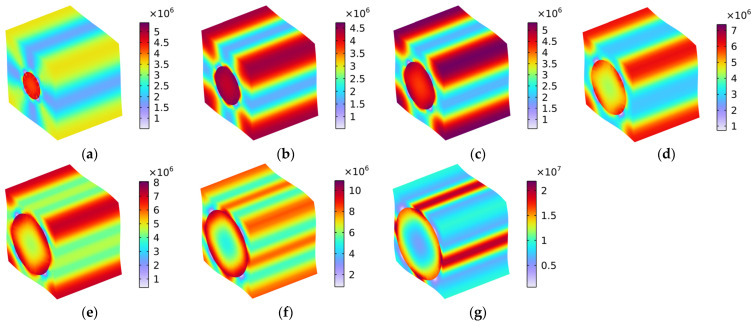
Distribution of Von Mises stress (N/m^2^) with various volume fractions under the condition of load in the y–z oblique plane. (**a**) Vf_c_ = 10%; (**b**) Vf_c_ = 20%; (**c**) Vf_c_ = 30%; (**d**) Vf_c_ = 40%; (**e**) Vf_c_ = 50%; (**f**) Vf_c_ = 60%; and (**g**) Vf_c_ = 70%.

**Figure 13 polymers-16-02120-f013:**
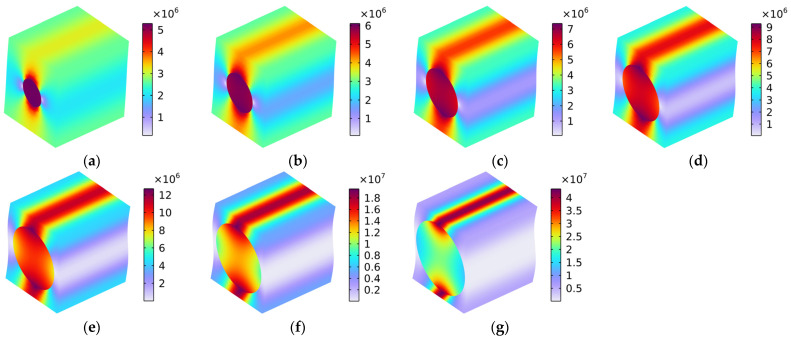
Distribution of Von Mises stress (N/m^2^) with various fiber volume fractions under the conditions of load in the x–z oblique plane. (**a**) Vf_c_ = 10%; (**b**) Vf_c_ = 20%; (**c**) Vf_c_ = 30%; (**d**) Vf_c_ = 40%; (**e**) Vf_c_ = 50%; (**f**) Vf_c_ = 60%; and (**g**) Vf_c_ = 70%.

**Figure 14 polymers-16-02120-f014:**
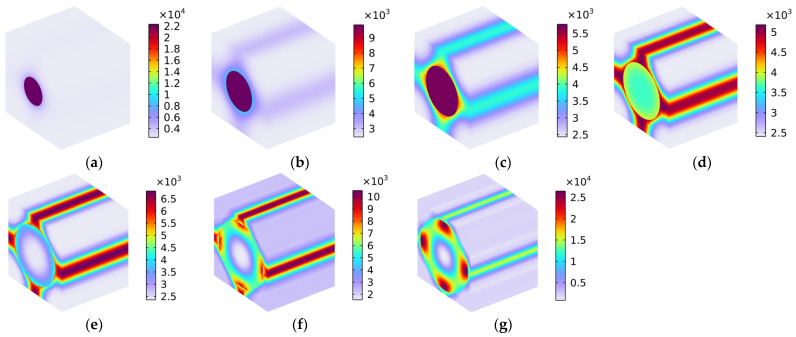
Distribution of Von Mises stress (N/m^2^) with various fiber volume fractions for the materials’ thermal properties. (**a**) Vf_c_ = 10%; (**b**) Vf_c_ = 20%; (**c**) Vf_c_ = 30%; (**d**) Vf_c_ = 40%; (**e**) Vf_c_ = 50%; (**f**) Vf_c_ = 60%; and (**g**) Vf_c_ = 70%.

**Table 1 polymers-16-02120-t001:** Selected parameters for the carbon fiber in the simplified micromechanical model.

Parameters	Symbol	Unit	Values
Young’s modulus	{E_xc_, E_yc_, E_zc_}	GPa	{230, 15, 15}
Shear modulus	{G_xyc_, G_yzc_, G_xzc_}	GPa	{15, 7, 15}
Thermal expansion coefficient	{α_xc_, α_yc_, α_zc_}	1/K	{−0.8 × 10^−6^, 25 × 10^−6^, 25 × 10^−6^}
Poisson’s ratio	{υ_xyc_, υ_yzc_, υ_xzc_}	1	{0.3, 0.08, 0.3}
Density	ρ_c_	Kg/m^3^	1800

**Table 2 polymers-16-02120-t002:** Selected parameters for rubber in the simplified micromechanical model.

Parameters	Symbol	Unit	Values
Young’s modulus	E_r_	MPa	4
Shear modulus	G_r_	MPa	1.36
Thermal expansion coefficient	α_r_	1/K	1 × 10^−4^
Poisson’s ratio	υ_r_	1	0.47
Density	ρ_r_	Kg/m^3^	920

**Table 3 polymers-16-02120-t003:** Various kinds of loads for cell periodicity in the simplified micromechanical model.

**Symbol**	Load 11	Load 22	Load 33	Load 12	Load 23	Load 13
**Loading direction**	x-axis	y-axis	z-axis	x–y oblique plane	y–z oblique plane	x–z oblique plane

**Table 4 polymers-16-02120-t004:** Performance comparison of the carbon fiber, rubber matrix, and composite material.

Vf_c_	0%	10%	20%	30%	40%	50%	60%	70%	100%
E_xx_ (GPa)	0.0004	23.0220	46.0217	69.0216	92.0217	115.0221	138.0232	161.0268	230
E_yy_ (MPa)	4	5.4078	6.1820	7.2254	8.7080	10.9916	15.0935	25.8701	15,000
G_xy_ (MPa)	1.36	1.6630	2.0411	2.5306	3.1982	4.1898	5.9053	10.1056	15,000
α_xx_ (1/K)	1 × 10^−4^	−4.9768 × 10^−7^	−5.5226 × 10^−7^	−5.7082 × 10^−7^	−5.8031 × 10^−7^	−5.8602 × 10^−7^	−5.8931 × 10^−7^	−5.8894 × 10^−7^	−8 × 10^−7^
α_yy_ (1/K)	1 × 10^−4^	5.7151 × 10^−4^	5.0573 × 10^−4^	4.4064 × 10^−4^	3.7611 × 10^−4^	3.1162 × 10^−4^	2.4510 × 10^−4^	1.6609 × 10^−4^	0.25 × 10^−4^

## Data Availability

Data are contained within the article.

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
