# Peer review of "Research on the Mechanical and Thermal Properties of Carbon-Fiber-Reinforced Rubber Based on a Finite Element Simulation"

_polymers, 2024, doi:10.3390/polym16152120_

Round 1
Reviewer 1 Report
Comments and Suggestions for Authors
Review report for polymers-3096279
In this manuscript, authors pointed out that to improve the research efficiency and decrease the experimental cost, the mechanical and thermal properties of the carbon fiber reinforced rubber were investigated by finite element simulation and theoretical modeling. They tried to construct the simplified micromechanical model through the repeatable unit cell with periodic boundary condition, and the corresponding theoretical models were built based on the rule of mixture (ROM), which could be treated as the mutual verification. The simulation results showed that along with the fiber volume fraction Vfc increasing from 10% to 70%, the longitudinal Young’s modulus, transversal Young’s modulus, in–plane shear modulus, longitudinal thermal expansion coefficient, and the transversal thermal expansion coefficient changed from the primary values to different ones. Authors mentioned that the influencing mechanism of Vfc to the properties of carbon fiber reinforced rubber was revealed by the distribution of Von Mises stress. They finally emphasized that this research was favorable to improve the performance of carbon fiber reinforced rubber and promote its application.
The subject is very interesting and beneficial for the applied purposes. The manuscript has been well organized. However, the concerns raised here must be addressed before manuscript acceptance. My comments are below.
1- Explain and highlight the novelties of this research work with respect to the previously published papers in the manuscript.
2- Please focus on title of manuscript with explanation of uniform literature review in the introduction section.
3- A mild revision is needed for English language of the manuscript.
4- It is strongly recommended comparing the simulation results with the experimental data.
5- It is necessary to mention and discuss on viscoelasticity of the used materials. Describe and discuss the viscoelastic phenomena, in general, in the introduction, or results and discussion sections. For this purpose, authors can read and cite the following [https://doi.org/10.1007/s10965-012-9897-2].
6- Please perform statistical analysis on the related results. For example, use the error bars.
7- Please explain the limitations of this study in the manuscript.
8- Abbreviations section should be added in the manuscript.
9- Please provide consistency between abstract and conclusion sections.
Comments on the Quality of English LanguageAuthor Response
Response to reviewer’s comments
General Comment:
In this manuscript, authors pointed out that to improve the research efficiency and decrease the experimental cost, the mechanical and thermal properties of the carbon fiber reinforced rubber were investigated by finite element simulation and theoretical modeling. They tried to construct the simplified micromechanical model through the repeatable unit cell with periodic boundary condition, and the corresponding theoretical models were built based on the rule of mixture (ROM), which could be treated as the mutual verification. The simulation results showed that along with the fiber volume fraction Vfc increasing from 10% to 70%, the longitudinal Young’s modulus, transversal Young’s modulus, in–plane shear modulus, longitudinal thermal expansion coefficient, and the transversal thermal expansion coefficient changed from the primary values to different ones. Authors mentioned that the influencing mechanism of Vfc to the properties of carbon fiber reinforced rubber was revealed by the distribution of Von Mises stress. They finally emphasized that this research was favorable to improve the performance of carbon fiber reinforced rubber and promote its application.
The subject is very interesting and beneficial for the applied purposes. The manuscript has been well organized. However, the concerns raised here must be addressed before manuscript acceptance. My comments are below.
Response:
Thank you very much for your helpful review and positive assessment to our manuscript. In this research We have revised the whole manuscript carefully according to your and other reviewers’ comments, and these corrections and modifications are highlighted in yellow in the revised manuscript.
- Explain and highlight the novelties of this research work with respect to the previously published papers in the manuscript.
Response:
Thank you very much for your valuable comment and helpful suggestion. according to your and the other reviewers’ comment, we have adjusted the presentations about the novelties of this research work in the sections of “1. Introduction” and “6. Conclusions” in the revised manuscript, and these corrections are highlighted in yellow.
- Please focus on title of manuscript with explanation of uniform literature review in the introduction section.
Response:
Thank you very much for your valuable comment and helpful suggestion. According to your comment, we have adjusted the presentations of the cited literatures in the sections of “1. Introduction”, and these modifications are highlighted in yellow in the revised manuscript. Meanwhile, limited by the length of paper required by this journal, not every literature was discussed, and the important achievements of some representative papers are pointed out.
- A mild revision is needed for English language of the manuscript.
Response:
Thank you very much for your valuable comment and helpful suggestion. We have polished the whole manuscript carefully to eliminate the possible spelling and grammar errors, which aims to make the manuscript more readable and easier to understand. These corrections are highlighted in yellow in the revised manuscript.
- It is strongly recommended comparing the simulation results with the experimental data.
Response:
Thank you very much for your valuable comment and helpful suggestion. As mentioned in the section of “6. Conclusions”, relative to the complex and time–consuming experimental test, the finite element simulation method could not only reduce the experimental cost, but also could visually show the influencing mechanism. Therefore, the mechanical and thermal properties of carbon fiber reinforced rubber had been investigated based on finite element simulation in this research, which aimed to study the influence rule and action mechanism. We agree with you that the experimental validation is an effective way, and we will attempt to conduct the corresponding experiments in future study. In this research, the finite element simulation result obtained by the simplified micromechanical model and the theoretical data achieved by the various ROM models were treated as the mutual verification.
- It is necessary to mention and discuss on viscoelasticity of the used materials. Describe and discuss the viscoelastic phenomena, in general, in the introduction, or results and discussion sections. For this purpose, authors can read and cite the following [https://doi.org/10.1007/s10965-012-9897-2].
Response:
Thank you very much for your valuable comment and helpful suggestion. We have found the recommended paper entitled “Molecular dynamics study of epoxy/clay nanocomposites: rheology and molecular confinement” at https://doi.org/10.1007/s10965-012-9897-2, and it is added in the reference list as [43]. Meanwhile, it was discussed in the paragraph 4 in section “1. Introduction”. Sodeifian et al. [43] had conducted the molecular dynamics study of epoxy/clay nano-composites, which proved that the chain relaxation process was slowed by the polymer–particle interactions. The modifications are highlighted in yellow in the revised manuscript.
- Sodeifian, G.; Nikooamal, H.R.; Yousefi, A.A. Molecular dynamics study of epoxy/clay nanocomposites: rheology and molecular confinement. J. Polym. Res. 2012, 19, 9897.
- Please perform statistical analysis on the related results. For example, use the error bars.
Response:
Thank you very much for your valuable comment and helpful suggestion. Taking the finite element simulation result as the reference value, the relative deviations for the various ROM models could be derived, as shown in the added Figure 7 in the revised manuscript. It could be judged from Figure 7 that for different properties, the deviations for various ROM models were not the same. For longitudinal Young’s modulus Exx, transversal Young’s modulus Eyy, in–plane shear modulus Gxy, longitudinal thermal expansion coefficient αxx, and transversal thermal expansion coefficient αyy, the corresponding ROM model with minimum deviation was Halpin–Tsai–Nielsen model, modified Voigt–Reuss model, Halpin–Tsai model, Chamis model, and Voigt–Reuss model respectively. Therefore, each ROM model had its suitable scenario, and appropriate ROM model should be selected to investigate the various property for the carbon fiber reinforced rubber. These modifications are highlighted in yellow in the revised manuscript.
- Please explain the limitations of this study in the manuscript.
Response:
Thank you very much for your valuable comment and helpful suggestion. As mentioned in your comment 4 “It is strongly recommended comparing the simulation results with the experimental data.”, the major limitation of this study is lack of experimental validation. The finite element simulation result obtained by the simplified micromechanical model and the theoretical data achieved by the various ROM models are treated as the mutual verification in this study, and the further experimental validation can make the whole research more reasonable. Therefore, as our response to your comment 4, We agree with you that the experimental validation is an effective way, and we will attempt to conduct the corresponding experiments in future study.
- Abbreviations section should be added in the manuscript.
Response:
Thank you very much for your valuable comment and helpful suggestion. According to the template of polymers, an abbreviation section is added at the end of the revised manuscript as the Appendix A. Meanwhile, the added Appendix A is highlighted in yellow in the revised manuscript.
- Please provide consistency between abstract and conclusion sections.
Response:
Thank you very much for your valuable comment and helpful suggestion. Because of the limitation of number of words in the abstract, a brief research process and main quantitative results are presented in abstract section, and the detailed results are listed in the conclusion sections. We agree with you that abstract and conclusion sections should keep consistency. Therefore, according to your helpful comment, we have modified and adjusted conclusion section through adding more quantitative data, which are consistent with the abstract section. These corrections are highlighted in yellow in the revised manuscript.
Reviewer 2 Report
Comments and Suggestions for Authors
The manuscript is subjected to some improvement before being accepted for publication:
1. Line # 114, the novelty of current study should be presented in a more clear way.
2. Line # 134, the Simplified Micromechanical Model was authors own or they have taken from other studies, if not then provide appropriate reference.
3. For sections 2 and 3, it is suggested to add the heading of materials and methods for better understanding.
4. Author should discuss and compare the properties of their material with other related materials in a tabular form.
Author Response
Response to reviewer’s comments
General Comment: The manuscript is subjected to some improvement before being accepted for publication.
Response:
Thank you very much for your kind review to our manuscript and constructive suggestion to our study. We have revised and corrected the whole manuscript carefully according to your and the other reviewers’ comments. The responses to your helpful comments are as follows.
- Line # 114, the novelty of current study should be presented in a more clear way.
Response:
Thank you very much for your valuable comment and helpful suggestion. The target of this research is to investigate the influence rule and reveal the action mechanism for mechanical and thermal properties of carbon fiber reinforced rubber based on the finite element simulation. This research method is not only applicable to the carbon fiber reinforced rubber in this study, but also can be used to study the properties of other fillers reinforced rubber products, and it can even provide effective reference to analyze other polymer composites. According to your helpful suggestion, we have corrected the presentations in the last paragraph of the section “1. Introduction”, and these corrections are highlighted in yellow in the revised manuscript.
- Line # 134, the Simplified Micromechanical Model was authors own or they have taken from other studies, if not then provide appropriate reference.
Response:
Thank you very much for your helpful review and valuable comment. It can be judged from the simplified micromechanical model for carbon fiber reinforced rubber in Figure 1 that it is quite simple, which is constructed by the repeatable unit cell (RUC) with the periodic boundary condition. Thus, we build this model by ourselves based on the RUC theory, which is a common method in the finite element simulation process.
- For sections 2 and 3, it is suggested to add the heading of materials and methods for better understanding.
Response:
Thank you very much for your helpful suggestion. According to your comments, the title of section 2 is corrected from “2. Simplified Micromechanical Model” to “2. Simplified Micromechanical Model of Carbon Fiber Reinforced Rubber”, and that of section 3 is corrected from “3. Theoretical Model” to “3. Theoretical ROM Models for Carbon Fiber Reinforced Rubber”. The relevant corrections are highlighted in yellow in the revised manuscript.
- Author should discuss and compare the properties of their material with other related materials in a tabular form.
Response:
Thank you very much for your valuable and meaningful suggestion. At the end of section “4. Results and Discussions”, comparisons of longitudinal Young’s modulus Exx, transversal Young’s modulus Eyy, in–plane shear modulus Gxy, longitudinal thermal expansion coefficient αxx, and the transversal thermal expansion coefficient αyy among carbon fiber, rubber matrix, and the composite material are shown in the added Table 4 in the revised manuscript. Here the Vfc=0% meant that it was pure rubber matrix, and Vfc=100% meant that it was pure carbon fiber. Meanwhile, the data for composite material was represented by the finite element simulation value, and those for the carbon fiber and rubber matrix were obtained from the Tables 1 and 2 respectively. it could be found that the reinforcement of carbon fiber to the composite material increased gradually along with the increase of its volume fraction Vfc, which proved effectiveness of carbon fiber filler. These modifications are highlighted in yellow in the revised manuscript.
Reviewer 3 Report
Comments and Suggestions for Authors
The introduction provides an overview of the literature relevant to the research. The micromechanical model is well described. The loading experiments correspond to real conditions and are of interest to scientists in the field of materials research. The structured conclusions make the results comprehensive. I recommend publishing it in its current form.
Author Response
Response to reviewer’s comments
General Comment:
The introduction provides an overview of the literature relevant to the research. The micromechanical model is well described. The loading experiments correspond to real conditions and are of interest to scientists in the field of materials research. The structured conclusions make the results comprehensive. I recommend publishing it in its current form.
Response:
Thank you very much for your positive assessment to our research results and manuscript. We have revised the whole manuscript carefully according to the editor’s and reviewers’ comments, and these corrections and modifications are highlighted in yellow in the revised manuscript.
Round 2
Reviewer 1 Report
Comments and Suggestions for Authors
Authors have tried to address the reviewers comments. Accordingly, it is now recommended publishing the revised manuscript at the present form.

Reviewer 2 Report
Comments and Suggestions for Authors
The authors have revised accordingly